# THE INVARIANCE STARVATION HYPOTHESIS

## ABSTRACT

Deep neural networks are known to learn and rely on spurious correlations during training, preventing them from being reliable and able to solve highly complex problems. While there exist many proposed solutions that overcome such reliance in different, tailored settings, current understanding regarding the formation of spurious correlations is limited. All proposed solutions with promising results assume that networks trained with empirical risk minimization will learn spurious correlations due to a preference for simpler features and that a solution to this problem requires further processing on the networks' learned representations or re-training on a modified dataset where the proportion of training data with spurious features is significantly lower. In this paper, we aim to form a better understanding regarding the formation of spurious correlations by performing a rigorous study regarding the role that data plays in the formation of spurious correlations. We show that in reasoning tasks with simple input samples, simply drawing more data from the same training distribution overcomes spurious correlations, even though we maintain the proportion of samples with spurious features. In other words, we find that if the network has enough data to encode the invariant function appropriately, it no longer relies on spurious features, regardless of its strength. We observe the same results in settings with more complex distributions with an intractable number of participating features, such as vision and language. However, we find that in such settings, drawing more samples from the training distribution while maintaining proportion can exacerbate spurious correlations at times, due to the introduction of new samples that are significantly different from samples in the original training set. Taking inspiration from reasoning tasks, we present an effective remedy to this problem to ensure that drawing more samples from the distribution always overcomes spurious correlations.

## 1 INTRODUCTION

Deep neural networks tend to form correlations between weakly predictive, spurious features and target labels during training. In practice, these networks often prefer these correlations over correlations formed between general, fully predictive invariant features and target labels. Thus, in the event of a distribution shift, where such spurious correlations may no longer hold, these networks begin to malfunction. Of the many factors that influence the degree of spurious feature reliance, the primary factor commonly discussed in literature is that of predictive power. In other words, what proportion of the training set contains the spurious feature and can, thus, enable the correct classification of during training. Intuitively, the larger the proportion, the greater the reliance of the network on the spurious feature. Thus, to prevent the formation of spurious correlations during training, common practice in deep learning encourages sampling training data from many different training environments, to reduce the proportion of samples in the training set that contain the spurious features and overcome selection bias (Arjovsky et al., 2019). Such sampling, however, is *expensive* and *not always feasible*.

In this paper, we attempt to answer the following questions: What if one continued to sample more data from the same training environments? In other words, what would happen if one were to maintain the proportion of the training data that contained the spurious feature but simply increased the amount of training data, drawn from the same distribution. What would be the impact on spurious feature reliance?

With answers to these questions, we describe the novel contributions and insights presented in this paper:

**Selection Bias does not form Spurious Correlations. Starving Networks of Sufficient Training Data Does.**  Through our experiments, we find that, although we maintain the proportion of samples that contain the spurious feature, simply drawing more data from the training distribution can overcome a model's reliance on spurious correlations and improve its robustness to distributional shifts. We show that if the network has **sufficient** data to encode the general, **invariant** function appropriately, it no longer learns and relies on spurious correlations present in the training data. Past works simply state that since deep neural networks are biased toward simpler predictive features, they are certain to learn and rely on spurious correlations. We refute this claim by showing that if the network is provided sufficient data to encode the invariant function well, it will no longer rely on simpler, weakly predictive spurious features present in the data.

**In Settings with Complex Distributions, Drawing More Samples from the Training Distribution can Exacerbate Spurious Correlations.**  In vision and language settings, drawing more samples from the training distribution can exacerbate spurious correlations. We find that this happens due to the introduction of new samples which contain general features that are not well represented in the original training set but also contain the spurious feature. If the general feature is not well represented and the network struggles to generalize to it, it forces the network to rely on spurious biases to minimize loss for that sample during training. We observe that in reasoning tasks, the input space is comprised of only a small number of objects, thereby making all samples, and by extension all general features, highly typical. Such typicality facilitates the easy reliance on general features over spurious ones. Hence, we use this knowledge to provide the model with samples that are generally well represented in the training distribution. This way, we overcome starvation and spurious correlations without running the risk of exacerbation.

**Present novel insights regarding the formation of spurious correlations across three domains: *reasoning, vision*, and *language*.**  We provide comprehensive empirical evaluation across three domains that are commonly studied in deep learning and show that our claims and observations are consistent across all three domains. That is, we show in all three domains that spurious correlations are formed due to deep networks being starved of sufficient data to be able to appropriately encode the general, invariant feature. Additionally, we utilize insights from reasoning tasks to better understand the nature of spurious correlations in more complex settings: vision and language. This direction, to the best of our knowledge, is novel.

## 2 RELATED WORK

**Spurious Correlations.**  Deep neural networks have a tendency to learn and rely on simpler, spurious cues that can aid in learning only a portion of a task (Arjovsky et al., 2019; Sagawa et al., 2020a;b; Ahmed et al., 2021; Liu et al., 2021; Zhang et al., 2022a; Kirichenko et al., 2022). This makes them brittle to distributional shifts as the spurious features that are relied upon may disappear or become correlated with a different task during testing. While there exists a large body of work that studies spurious correlations, combating this problem remains an open problem. Most proposed techniques that mitigate spurious correlations require domain knowledge, are effective only in specific settings or negatively impact test accuracy. In this work, by studying reasoning tasks, we provide novel insights fundamental to deep neural network training that can help solve the spurious correlations problem.

**Simplicity Bias** Past work has shown that deep neural networks are biased towards simpler features, where in the presence of two fully predictive features, a model will rely only on the simpler feature and fully ignore the complex feature (Shah et al., 2020). Recent works have shown that even in settings where the simpler feature is not fully predictive of the task, the model still relies strongly on these features (Geirhos et al., 2020; Kirichenko et al., 2022). Such simplicity bias is considered as the primary reason behind the formation of spurious correlations. In this work, we form a better understanding regarding the role of simplicity bias in the formation of spurious correlations. We show that if the network has enough data to learn the general feature appropriately, it no longer relies on simpler, weakly predictive spurious features. Our observations disagree with the current notion which simply assumes that spurious correlations will always be formed in the presence of simpler, predictive, spurious features.

**Reasoning in Deep Learning.**  There has been an increasing interest in deep neural networks solving reasoning tasks that are either mathematical, visual, physical, or algorithmic in nature (Saxton et al., 2019; Bakhtin et al., 2019; Velickovic et al., 2022). Most existing works show that deep

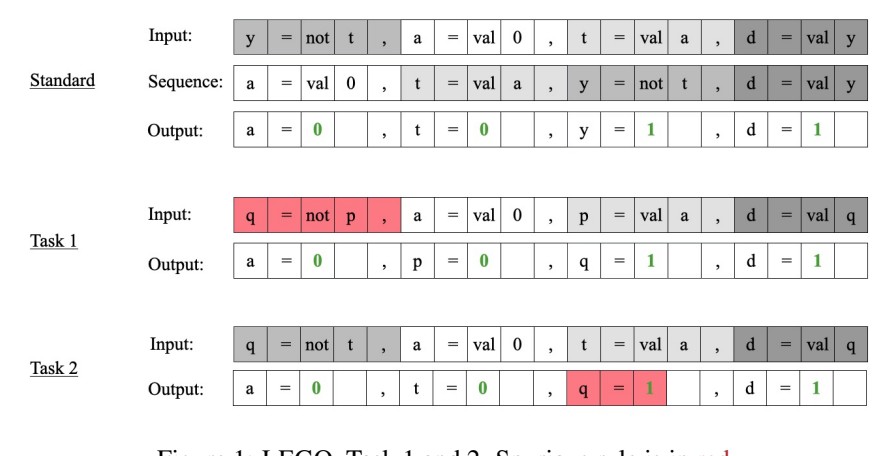

Figure 1: LEGO, Task 1 and 2. Spurious rule is in red.

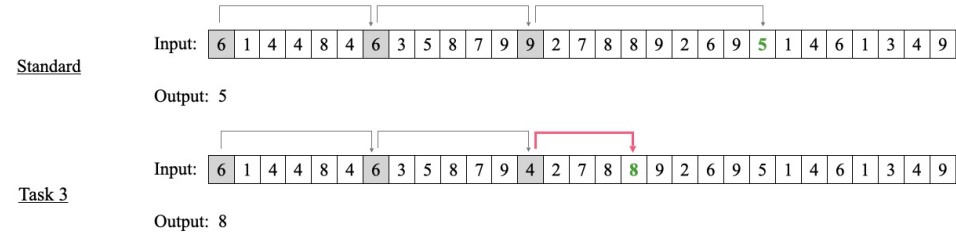

Figure 2: Modified Pointer Value Retrieval, Task 3. Spurious rule is in red.

neural networks are capable of understanding and solving reasoning tasks. To better enable deep neural networks to understand reasoning tasks, recent works also recommend making architectural and optimization changes to the learning process Ahmed et al. (2022; 2023); Zhang et al. (2023a); Marconato et al. (2023a); Giunchiglia et al. (2022). In this paper, we study specific behaviors of transformer-based models on sequential reasoning tasks. A growing body of work has shown that deep neural networks have a tendency to converge to short-cut solutions to solve reasoning tasks (Liu et al., 2023; Abbe et al., 2023). These short-cut solutions, at times, prevent deep networks from generalizing to different or unseen settings and domains (Zhang et al., 2022b; Zhou et al., 2024). Zhang et al. (2023b) find that deep networks rely on attributes of the training set to make predictions. A good example of this is the length generalization problem where a network that has been trained on a shorter sequential reasoning task fails to generalize to samples with longer sequences that require the same set of rules (Zhang et al., 2022b; Zhou et al., 2024). For another instance, Marconato et al. (2023b) show that NeSy predictors can, at times, misunderstand concepts in the training data.

## 3 DEEP NEURAL NETWORKS ARE STARVED OF INVARIANT INFORMATION

We begin our study with reasoning tasks commonly studied in literature that are based on learning rules that operate on a small set of features. In particular, we study the Learning Equality and Group Operations (LEGO) Zhang et al. (2022b) and Pointer Value Retrieval (PVR) Zhang et al. (2021) tasks. Through our experiments, we highlight a *novel failure mode* of deep networks on reasoning tasks, where a network learns the invariant rule but also learns spurious rules due to minor imperfections that are inherent to real-world data. We detail our experimental set-up below.

### 3.1 LEARNING EQUALITY AND GROUP OPERATIONS (LEGO)

The LEGO task is a sequential reasoning task where the input is a sequence of variable assignments and operations on these variables (`Input` in Fig. 1). The solution for the LEGO task takes the form of a loop where values for variables are resolved one at a time and every new variable encountered is resolved using the previously resolved variable (`Sequence` and `Output` in Fig. 1). While Zhang

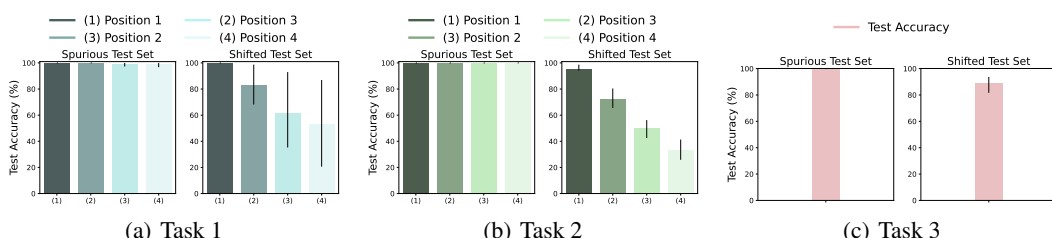

(a) Task 1          (b) Task 2          (c) Task 3

Figure 3: Test accuracy shown on Test Set with the same distribution as Train Set and Spurious Test Set. (Every sample follows the invariant rule but breaks the spurious rule.)

et al. (2022b) show that an encoder-only transformer model can learn the invariant rule, they only train on a dataset that is ideal and unrealistic. They curate such a set by sampling variables, operations, values, and positions uniformly. We show that once we step away from this ideal environment, the network's generalizability suffers.

## 3.2 POINTER VALUE RETRIEVAL (PVR)

In the original PVR task, each training instance consists of a sequence of numbers, where the first number in the sequence behaves as a pointer which points to the label of the training instance, which is the number in the sequence that is indexed by the pointer. An example is shown in `Standard` in Fig. 2. As in the LEGO task, the original PVR task sequences are sampled uniformly, thereby creating an ideal environment for invariant rule learning.

## 3.3 TRAINING DETAILS

For both tasks, we use a pre-trained BERT, an encoder-only transformer model, for training and a pre-trained BERT tokenizer for tokenization. Consistent with the original implementation for LEGO, we use cross entropy loss averaged over the 4 clauses belonging to each sample during training. For PVR, we use standard cross entropy loss. The network is trained for 100 epochs with a batch size of 1,000 samples and optimized with Adam using a learning rate 5e-5 and cosine learning rate schedule with Tmax = 100. In Task 1, Task 2 and Task 3, we maintain the training set size as 30,000, 4,000 and 37,500, respectively.

## 3.4 TASK DEFINITIONS AND OBSERVATIONS

In this paper, we curate and study the impact of training on the following three tasks:

**Task 1:** For Task 1, based on LEGO, we create a dataset where the variables $q$ and $p$ always occur together and $q$ is equal to the *negation* of $p$, as shown in `Task 1` in Fig. 1. Note that not all samples contain all literals and thus, this rule is only enforced in a portion (majority) of all samples.

**Observation:** On testing on a uniform test set, we observe that almost all misclassified samples are those where $p$ and $q$ have the *same* ground truth value or if $q$ is preceded by any literal in the loop that contains the same ground truth value. This results in poor testing accuracy, as shown in Fig. 3(a).

**Task 2:** We create a dataset based on LEGO where the variable $q$ always has the value 1, as shown in `Task 2` in Fig. 1. Here, the variable $q$ occurs in even fewer samples than it would have been a part of if one were to create a dataset by sampling uniformly.

**Observation:** Again, we observe that all misclassified samples are those that contain a $q$ but its ground truth value is 0, resulting in poor testing accuracy (Fig. 3(b)).

In both tasks, we observe that the network makes a mistake only once it encounters the variables participating in the broken spurious rule. Until then, all predicted values are correct. Interestingly, once a network makes a mistake, for most of the samples, every subsequent predicted value is

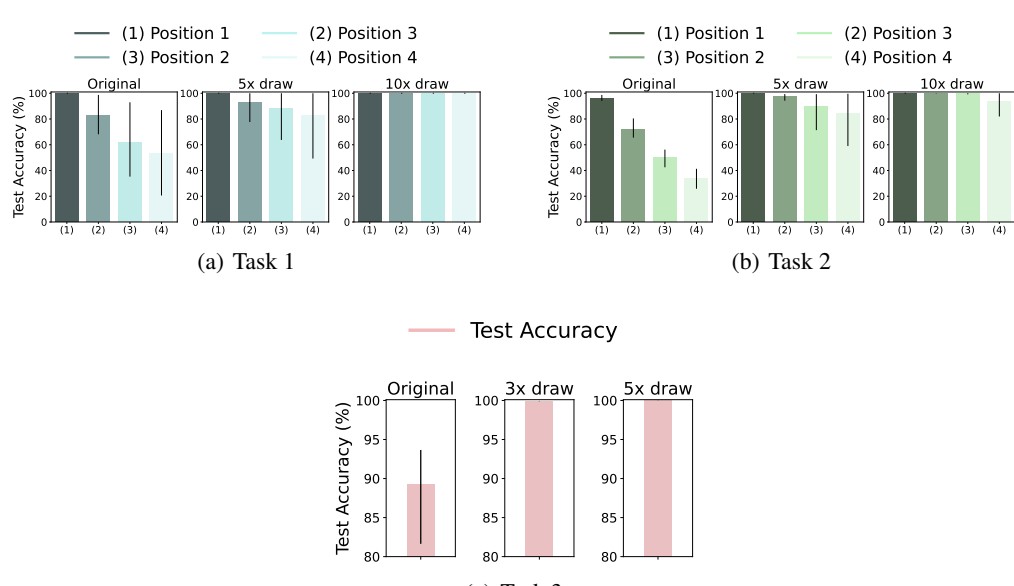

(a) Task 1            (b) Task 2

(c) Task 3

Figure 4: Drawing more samples from the same training distribution can mitigate spurious rule reliance. Note that we maintain the proportion of samples that encode the spurious rules.

incorrect (Figs. 3(a) and 3(b)), implying that it goes back to relying on the invariant rule after relying on the spurious rule. This is because a variable can take only one of two values, and thus, if subsequent incorrect predictions were caused by randomness, not all predictions would be incorrect, but only about 50% of the predictions would be incorrect.

**Task 3:** We create small modifications to the original PVR task. First, we make the task more complex by introducing additional steps to the task, where the index is computed in three steps instead of one. The label is computed by making 3 hops and the size of each hop is determined by the number at the position the hop lands on. The number after all hops is the label for that instance (Fig. 2). Next, we modify the data generation process such that in a significant portion of the dataset, the label is always equal to (number at last hop $+3)\%9+1$, as shown in `Task 3` in Fig. 2.

**Observation:** On testing on a uniform test set, we observe that most misclassifications occur when the samples do not encode the spurious rule, resulting in poor test accuracy (Fig. 3(c)).

## 4 OVERCOMING SPURIOUS CORRELATIONS IN REASONING TASKS BY DRAWING MORE SAMPLES FROM THE SAME DISTRIBUTION

In all three tasks, we observe that by simply sampling more data from the same training distribution with spurious features, the network overcomes reliance on the spurious feature present in the training data, attaining perfect accuracies on datasets that break the spurious rule but still perfectly encode the invariant rule as shown in Fig. 4.

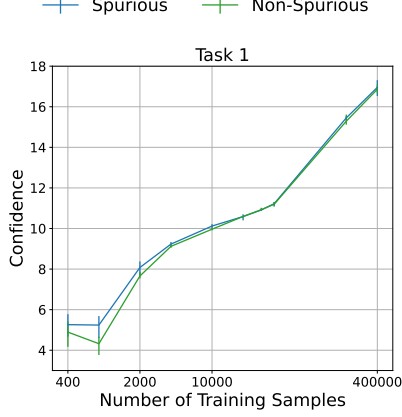

Figure 5: In the low data regime, the network makes use of the spurious rules to increase its margin (*i.e.*, confidence). In the high data regime, it no longer needs to rely on them. Confidence values are computed on the logit values of the binary classification task in Task 1.

Note that we simply sample more data from the same distribution and thus, maintain the proportion of samples that encode the spurious rule. The network still overcomes the reliance on spurious correlations. In other words, if we provide enough invariant information to a network such that it is able to encode the invariant, fully predictive function properly, it will *learn to ignore spurious signals* irrespective of their strength.

We verify our claim by observing differences in confidence/margin when a network is trained on a dataset that encodes both the invariant and the spurious rule against one that is trained on a dataset that only encodes the invariant rule. We see that in the low data regime, the network is unable to encode the invariant function properly and thus, it utilizes the spurious rule to increase its confidence during training as shown in Fig. 5. However, as we scale up the size of the training dataset and the network is able to classify training instances with sufficient confidence, the network no longer relies on spurious cues to increase its confidence.

Based on these observations, we ask the following question: Are deep neural networks simply starved of invariant information? Can supplying sufficient invariant information overcome spurious correlations, regardless of the strength of the spurious signal? We examine these questions in settings with complex data distributions that have abundant representative data, vision and language.

## 5 DRAWING MORE SAMPLES CAN EXACERBATE SPURIOUS CORRELATIONS IN COMPLEX DISTRIBUTIONS

In this section, we verify if our findings from reasoning tasks apply to popular benchmarks studied in literature that exhibit more complex distributions with abundant representative data.

### 5.1 EXPERIMENTAL SET-UP

We detail our experimental set-up below:

- **CelebA (Liu et al., 2015).** We create a gender classification task using the CelebA dataset, where a small fraction of the Male samples contain eyeglasses, which is the spurious feature in our setting. We estimate the degree of spurious feature reliance by measuring the number of Female samples with eyeglasses that are misclassified during testing. The lower the accuracy for Female samples with eyeglasses, the greater the degree of spurious feature reliance.

- **MultiNLI (Williams et al., 2018).** Inspired by experiments in Sagawa et al. (2020a), we create a three-class classification task with the target labels - entailed by, neutral with, or contradicts. In our experimental setting, the contradicts class contains a few samples with negation words, which is the spurious feature in our setting. We estimate the degree of spurious feature reliance by measuring the number of samples belonging to the `neutral with` or `entailed by` classes that contain negation words that are misclassified during testing. The lower the accuracy for these samples, the greater the degree of spurious feature reliance.

Note that our evaluation is consistent with current practice, where we estimate the degree of spurious feature reliance by measuring Worst-Group Accuracy (WGA), which computes the accuracy of test samples that contain the spurious feature associated with the other class during training. Additionally, consistent with all works that study spurious correlations, we perform hyperparameter tuning using a validation split to optimize for worst group accuracy (Sagawa et al., 2020a;b; Liu et al., 2021; Zhang et al., 2022a; Ahmed et al., 2021; Kirichenko et al., 2022).

### 5.2 TRAINING DETAILS

In the CelebA setting, we use an ImageNet pre-trained ResNet-50 model that we train for 25 epochs, optimized using SGD with a static learning rate 1e-3, weight decay 1e-4, and batch size 64. In the MulitNLI setting, we use a pre-trained BERT model that we train for 20 epochs, optimized using AdamW with a linearly decaying starting learning rate 2e-5 and a batch size of 32. In the CelebA setting, we maintain the training set size as 1,000, while in the MultiNLI setting we maintain the

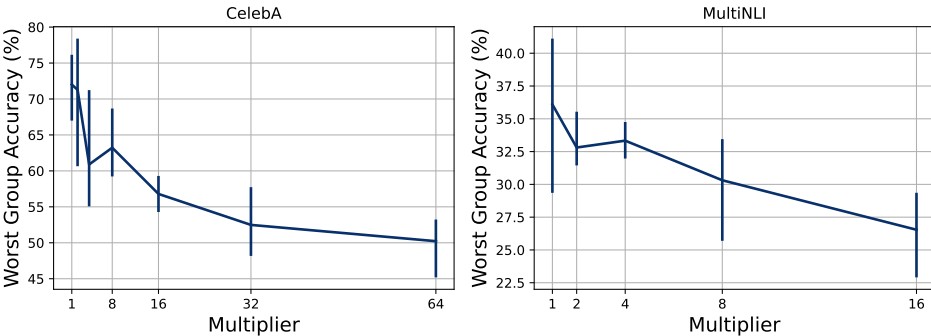

Figure 6: In settings with more complex data distributions such as vision and language, drawing more data from the distribution can hurt Worst Group Accuracy (WGA), implying an exacerbation in spurious correlations.

training set size as 6,000. In both settings, we start with this fixed number of samples per group and repeatedly double that amount. Each time we increase the dataset size, we measure and plot the worst group accuracy, as shown in Fig. 6.

## 5.3 OBSERVATIONS

Through our experiments, we find that in both settings, ***simply drawing more samples from the same distribution exacerbates spurious correlations.*** Note that we,

- Maintain the proportion of spurious vs. invariant information. In other words, we maintain the proportion of samples which contain the spurious feature in the training set.

- Introduce only a small number of samples with the spurious feature, and thus, each time we double the training set size while maintaining the proportion of spurious-to-invariant information, the number of new samples without spurious feature is a lot greater than the number of new samples with the spurious feature.

Despite this, drawing more samples from the same training distribution exacerbates spurious correlations in settings with complex distributions. This is in contrast to the experimental results shown for reasoning tasks (please refer to Fig. 4), where drawing more samples from the training distribution overcomes spurious correlations.

**RANDOM SAMPLING CAN INTRODUCE SAMPLES THAT CONTRIBUTE STRONGLY TO SPURIOUS FEATURE RELIANCE IN COMPLEX DISTRIBUTIONS**

In the reasoning settings studied above, all tasks have training distributions where the input space is comprised of only a small number of features. In such a setting, it is unlikely that the network will encounter samples that are atypical or poorly represented in the training distribution. In other words, training samples are highly similar to each other and this remains true regardless of the number of additional samples that we add to this training pool. This is in contrast to tasks in vision and language that have more complex distributions with abundant representative data, comprised of general features that exhibit high variance from sample to sample. Sub-sampling from these distributions may introduce multiple samples with core, invariant features that are not well represented in the original training dataset and are even harder for a network to understand. We observe that the maximum error ($\|p(w, x) - y\|_2$) a model attains on a sample during training in the reasoning tasks is almost the same as the minimum error a model attains on a sample (3.1003e-05 and 1.4378e-05, respectively). This is in sharp contrast to the CelebA setting studied, where the maximum error is far greater than the mininum error a model attains on a sample (2.047e-1 and 2.8412e-11, respectively).

In such settings with complex distributions, randomly sampling more data from the training distribution, as was effectively done in reasoning tasks, can introduce many samples that are difficult for a network to understand, thereby forcing it to rely strongly on spurious biases present in these samples when minimizing training loss.

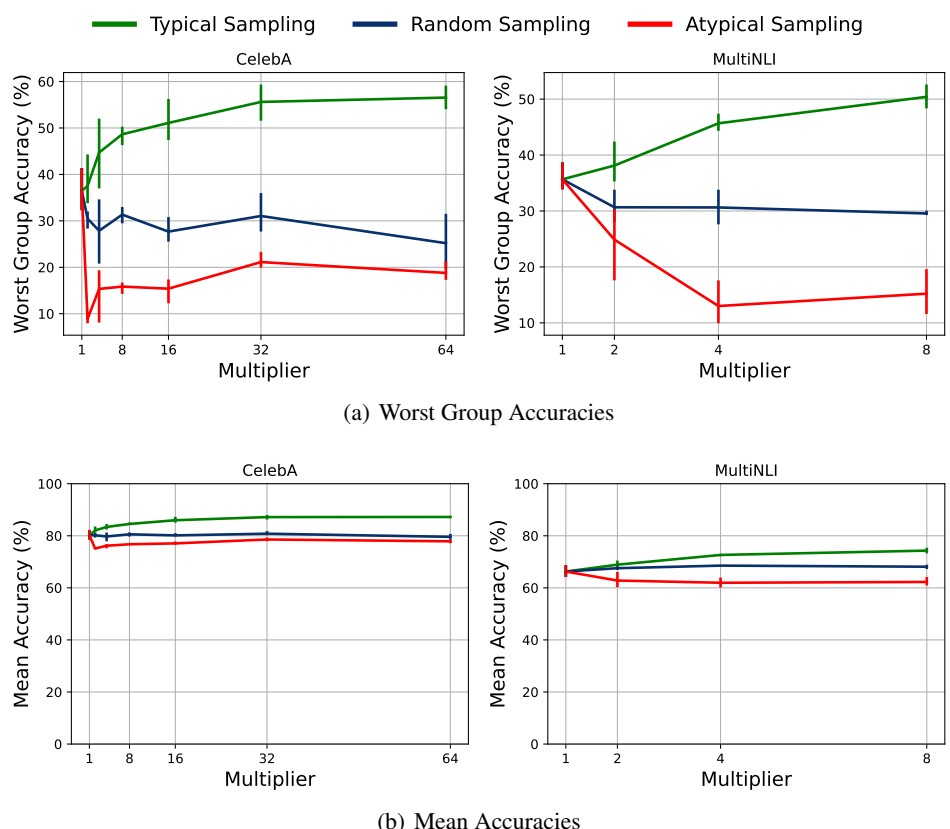

(a) Worst Group Accuracies

(b) Mean Accuracies

Figure 7: Drawing typical samples overcomes spurious correlations. Drawing atypical samples significantly exacerbates spurious correlations. Note that we maintain the proportion of samples containing the spurious feature in all three, Typical, Random and Atypical.

As such, in these settings, it is important to carefully draw from the training distribution to overcome invariance starvation. More specifically, to overcome starvation in such settings, it is important to draw samples with clear invariant information that represents the initial sub-sampled training set well.

## 5.4 OVERCOMING STARVATION IN COMPLEX DISTRIBUTIONS

To overcome starvation in settings with complex distributions, we aim to provide the network with additional training data that contains general features that are well represented or typical in the original training data.

To identify such samples in the training distribution, we first train the network on the entire (available) training distribution for that task. We then compute the training error early in training. This method of estimating which samples are well represented in the training distribution is inspired by Paul et al. (2021) that assigns a similar score for each sample in a dataset to determine which samples one must prune for efficient training. They claim that samples that have a lower error early during training are redundant/typical. Alternatively, samples that have a higher error during training are atypical.

We follow a similar method of estimating if a sample is well represented in a training distribution. Those samples that have a low error early in training are samples that are easy to learn and are typical and well represented in the training distribution. We compute such scores after the $10^{th}$ and $5^{th}$ epochs in the CelebA and MultiNLI settings, respectively. Note that in our experimental settings, the spurious features exhibit significantly lower variances than their general counterparts and thus, loss during training is primarily determined by the core features. So for instance, in the MultiNLI

setting, the spurious features takes the form of a couple of negation words that are exactly the same throughout the dataset.

In Fig. 7, we show that simply drawing more samples with typical or easy to learn general features can overcome spurious correlations while maintaining the proportion of spurious samples in the training set. Interestingly, providing new samples which contain core features that are poorly understood and are not well represented in the training distribution (atypical samples) exacerbates spurious correlations far more than when one were to simply sample randomly. Note that to show the efficacy of our technique in the CelebA, we start with a training set where the proportion of samples containing the spurious feature is greater than in the set-up shown in Section 5.1. This is because, in the original setting, the worst group accuracy is initially very high. However, for completeness, we still show the impact of random sampling in the new setting.

## 6 CONCLUSION

**Summary.** We show that in practice, deep neural networks are often starved of invariant information, making them highly sensitive to spurious features present in training data. We show that in reasoning tasks, while maintaining the proportion of spurious samples of the original training distribution, simply drawing more samples from the training distribution can overcome spurious correlations. We find that if the model has sufficient invariant information, the model does not rely on spurious features even if the proportion of spurious information is maintained. Surprisingly, in tasks with more complex distributions with abundant representative data, drawing more samples from the training distribution exacerbates spurious correlations. We find that this happens due to the existence of samples that are atypical or difficult for a network to generalize to, forcing them to rely on weakly predictive spurious features present in these samples. Finally, we show that in such settings, if one carefully draws samples with easier invariant features from the training distribution, one can overcome invariance starvation and mitigate spurious correlations.

**Impact.**

- **Deep neural networks do not always prefer simpler predictive features.** Current notion regarding the nature and formation of spurious correlations assumes that deep neural networks trained using empirical risk minimization will always form and rely on spurious correlations as deep neural networks are biased towards simpler features. Shah et al. (2020) show that in the presence of two fully predictive features, a model will choose to rely only on the simpler feature and completely ignore the complex feature. More recent works show that even if the simpler feature is not fully predictive of the task, a network will still rely strongly on these features and often ignore the general, invariant feature (Kirichenko et al., 2022). As such, it is assumed that spurious correlations will always be learned in the presence of simpler, partly predictive spurious features and that to overcome this problem, one must alter the network's biased representations or reduce the predictive power of the spurious feature in the dataset.

  Our paper is the *first* to show that models are not always biased to simpler features. We show that if the network is provided with enough data to encode the general, invariant function well, it will ignore the simpler, weakly predictive spurious features.

- **Sampling from multiple different training environments is not necessary.** Past works state that to avoid the formation of spurious correlations during training, one must sample from multiple different training environments or up-weight samples from environments that do not contain strong spurious cues (Arjovsky et al., 2019; Sagawa et al., 2020a; Liu et al., 2021). This ensures that the proportion of samples in the training dataset that contain strong spurious cues is reduced. In this paper, we show that one can continue to draw from the same training environments and maintain the same proportion of samples with spurious features but still overcome spurious correlations by simply providing more data.

- **More data can exacerbate spurious correlations and hurt generalizability.** Current notion in deep learning assumes that more data is almost always beneficial for generalizability. Through our experiments, we show that this is not always true as more data can exacerbate spurious correlations and that one must be careful when using more data to train their models.

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
