# OpenReview forum: "The Invariance Starvation Hypothesis"
_ICLR.cc/2025/Conference — ICLR 2025 Conference Withdrawn Submission_

### Official Review · Reviewer_WhYE · 2024-10-31

**Soundness:** 1
**Presentation:** 2
**Contribution:** 2
**Rating:** 3
**Confidence:** 4

**Summary:**

This paper aims to shed some light on the spurious correlation bias problem of neural networks. The authors study the effect of data on the formation of these correlations by investigating the behavior of models in reasoning and more complex tasks (such as vision and language). Authors are skeptic about the common statement that “neural networks are biased toward simpler predictive features” as a result of Empirical Risk Minimization. They pose two questions: What if one continued to sample more data from the training environments? What would be the impact on spurious feature reliance? Their results for reasoning tasks show that despite keeping the proportion of instances supporting spurious rules, the network will eventually learn to encode the invariant function with more data (drawn from the same distribution). That is: selection bias does not form spurious correlations, starving networks of sufficient training data does.
On the other hand, more complex settings behave differently due to the complexity of the data distribution, which leads to the addition of new samples that are significantly different from the others even when drawing from the same distribution, and thus exacerbating spurious correlations.

**Strengths:**

- The problem at hand is important and not well tackled yet. Many real world situations contain spurious correlations and disentangling them from the invariant features is a difficult task.
- Good related work section.

**Weaknesses:**

- The main claim of this work, presented in line 471, "Our paper is the first to show that models are not always biased to simpler features. We show that if the network is provided with enough data to encode the general, invariant function well, it will ignore the simpler, weakly predictive spurious features." is only sustained with a simple, synthetic experiment. It consists of a reasoning problem (a deterministic problem) but it is immediately invalidated once brought to a more complex distribution (a stochastic problem) on a Computer Vision and NLI problem. The paper states a strong claim on simple, deterministic settings, and later invalidates said claim on stochastic problems, but still reaffirms the claim as a novel contribution. The argument is clearly flawed and requires more work. Why is it not working on stochastic processes? Can we really "rectify" this issue by employing typical samples as suggested on section 5.4? The true value of this paper is on these questions but they are not sufficiently developed.
- This final claim on section 5.4 is not sufficiently proved either, as it is only evaluated on two datasets for one particular spurious feature each. Besides, no error bars are presented to determine if the observed effects are truly significant. Further testing of this claim is required to ensure the validity of the claim—can we extend the initial claim to stochastic, more complex and realistic settings?—, but as it is, the paper does not offer an answer.
- Experiment figures are not clearly explained. Figure 3: what is the shifted dataset? What are the different positions in a) and b)? Even if it can be understood by context, it is not clearly stated.
- The presentation is a bit difficult to follow because of the atypical titles in sections. I would recommend to use succinct titles for sections instead.

**Questions:**

- Have the authors tried some simpler datasets for the complex tasks such as vision? For example, the Waterbirds vs. Landbirds is a simple image classification problem where spurious correlations are strong. This would offer a smaller experiment on stochastic problems that support the claim.
- I would ask to include global accuracies after the modified training schedule in section 5.4 to attest that overall performance is not impacted by this change.
- I would ask to include more detail in how “typical samples” are prioritized during training in section 5.4.
- This paper does not discuss the causal perspective in dealing with problems of distribution shift, like these spurious features. I would include a reference on this topic and possibly a short discussion on this alternative approach. See [1].

[1] Schölkopf, Bernhard, et al. "Toward causal representation learning." Proceedings of the IEEE 109.5 (2021): 612-634.

---

### Official Review · Reviewer_YxDy · 2024-11-01

**Soundness:** 1
**Presentation:** 1
**Contribution:** 2
**Rating:** 3
**Confidence:** 4

**Summary:**

This empirical paper studies the over-reliance of deep learning models on spurious correlations. Given that some prior work has suggested that training while sampling from diverse environments can be helpful for learning invariant features, the authors explore to what extent sampling more data from the original training distribution can help.

The authors empirically evaluate model performance (ResNet, BERT) on four datasets: two synthetic reasoning benchmarks, gender recognition with CelebA, and natural language inference with MNLI. They present results suggesting that for reasoning tasks, collecting more samples from the training distribution is sufficient to reduce the reliance on spurious correlations. The authors argue this is a result of the simplicity of reasoning tasks. In contrast, they suggest that for vision and language tasks, sampling more data from the training distribution exacerbates the use of spurious correlations, because the invariant are poorly-represented, forcing the model to still use the spurious features.

Finally, the authors use this insight to mitigate spurious correlations, finding that when upweighting easy samples, increasing training samples can indeed be helpful for the vision and language tasks considered.

**Strengths:**

- At its core, this paper asks an important question. Can models be dissuaded from learning spurious correlations if we simply present an increasing number of samples from the training distribution?
- The finding that presenting easy (according to early-stage training loss) samples can reduce over-reliance on spurious correlations is interesting, and could have practical utility.

**Weaknesses:**

- This work suffers from significant issues in its approach, evaluations, and conclusions. Overall, I do not find the results presented to justify the large claims scattered throughout the paper.
- In its current form, the paper argues that our conception of why models use spurious correlations, rather than learning invariant features, is incorrect, principally relying on results from simple reasoning tasks. However, given that the results on CelebA and MultiNLI are in line with current knowledge (i.e., scaling training samples is not helpful for reducing worst-group error), then I am more inclined to ask why the reasoning results differ. Do they suggest an error, an edge case, or something fundamentally different? At present, without more systematic evaluation, it is impossible to tell.
- The authors attempt to suggest that reasoning tasks differ because of their simplicity (e.g., "We observe that in reasoning tasks, the input space is comprised of only a small number of objects, thereby making all samples, and by extension all general features, highly typical"; lines 069--071). This is highly likely to be a property of the synthetic tasks (LEGO, PVR) tasks considered, rather than a general property of reasoning. Indeed, reasoning itself is often a core component of solving other tasks in language and vision, such as in performing natural language inference (e.g., MNLI). If, as the authors suggest, that increasing the number of samples is sufficient to overcome spurious correlations in reasoning tasks, then why does this approach not also help with MNLI?
- More broadly, to support such big claims, the evaluations are too limited. If the authors wish to make general claims about differences between reasoning-based tasks and tasks from other domains, then they will need to evaluate multiple benchmarks in each category to prove the generality of their results. I would suggest, at the very least, expanding their analysis to consider other well-studied benchmarks in the worst-group optimization space, such as Waterbirds for vision and CivilComments for language.
- Fig 7a (blue line) shows almost no effect of random sampling on worst-group accuracy when increasing scale. This directly contradicts the paper's claim about model reliance on spurious correlations being exacerbated with scale on vision and language tasks.
- The claim "Selection Bias does not form Spurious Correlations" is not supported by the evidence. At best, the authors could claim that "selection bias is not the sole cause of spurious correlations."
- In section 5.3, the authors state that "the number of new samples without spurious feature is a lot greater than the number of new samples with the spurious feature" (lines 353--354). This statement is misleading, as it ignores the fact that absence of the spuriously-correlated feature can also be predictive of the target label in the tasks considered.
- The paper contradicts itself in the abstract and the introduction. On the one hand, the authors wish to indicate dichotomized behaviour between simple/complex reasoning/non-reasoning tasks. Yet, at the same time, the authors wish to suggest their results are consistent across domains. For example, lines 025--026 state "We observe the same results in settings ... such as vision and language," while lines 027--028 state "we find that in such settings, drawing more samples from the training distribution ... can exacerbate spurious correlations." Both of these cannot simultaneously be true.
- The paper uses inaccurate language such as "formation of spurious correlations" and "selection bias does not form spurious correlations". Spurious correlations are a property of the data; the subject of this work is whether models learn to use them. There are also  grammatical errors and unnecessary capitalizations (e.g. lines 054-055) throughout.
- Minor: Predictive power does not simply refer to which proportion of the training set contains the spurious feature (lines 040-041). Trivially, one could have a feature present in 99% of samples that is highly noisy, and therefore has low predictivity.
- Minor: Past works do not "simply state" that neural networks are biased towards using simpler spurious correlations - they present empirical and theoretical evidence that this is the case. The authors should avoid misrepresenting the state of the literature.

**Questions:**

1. Where is the evidence to support the claims in the subsection on lines 361--413? I only see a comparison of test error on line 372 and 374.
2. Have the authors analyzed or visualized which samples are actually presented during "typical" sampling, given that "typical" samples are defined as those with low loss early in training? My suspicion would be that these samples do not exhibit the spurious correlation. Arguably, by biasing the training sampling process you are effectively drawing from a new distribution, where the spurious correlation may not hold. This is equivalent to sampling from another environment.

---

### Official Review · Reviewer_oWAP · 2024-11-03

**Soundness:** 2
**Presentation:** 3
**Contribution:** 2
**Rating:** 3
**Confidence:** 4

**Summary:**

This paper examines how spurious correlations affect deep neural networks and explores the role of data in mitigating their effects. The authors find that in simple reasoning tasks, increasing the data from the same distribution—while keeping the proportion of spurious features—helps the network learn invariant functions, thus reducing reliance on spurious correlations. However, in complex tasks like vision and language, simply adding more data can sometimes worsen spurious correlations due to sample diversity. To address this, the paper proposes a solution that helps overcome spurious correlations by sampling from data only with low early training errors. This approach offers insights into improving the reliability of neural networks in complex settings.

**Strengths:**

- This paper offers an interesting view on why neural networks are prone to spurious correlation. In short, there are two main reasons: one is that the sample size (of the training data) is too small and the other is that the sample is too noisy. In either case, it is difficult for the model to capture the invariance in data. This is interesting because it suggests that both data quantity and quality and important for training a robust model.
- The paper is clearly written and easy to follow.

**Weaknesses:**

- The results are not really surprising. It is known that over-parameterization can exacerbate spurious correlation [1]. In other words, having more data, which reduces over-parameterization, can help overcome spurious correlation. There is no real difference between having more/fewer parameters (as in [1]) and having fewer/more data (as in this work). Moreover, [1] has also shown that the opposite can happen too, i.e., fewer parameters (or more data) sometimes exacerbate spurious correlation. This is similar to the second main empirical observation of the paper presented in Section 5.
- The explanation for why more data contribute to spurious correlation reliance is not well justified. The paper attributes this phenomenon to the complexity of the data distribution but lacks good arguments to back it up. The explanation essentially says that data sampled from complex distributions are of high variance and thus it is difficult for a network to capture the invariant features from the data. However, this still does not explain why adding more data makes it even harder for the network to learn correctly.
- The empirical evidence presented in the paper is not compelling. First of all, one would expect more extensive and rigorous experiments to support general claims like “invariance starvation hypothesis”. The experiments only involve two synthetic reasoning tasks, one vision task, and one language task. They are conducted on only a very limited amount of different architectures and hyperparameter settings. Second, there are too many variables between the two sets of experiments (reasoning vs. vision & language) to clearly demonstrate the impact of the complexity of the data distribution.
- Many experiment details are unclear. Were all the networks trained until convergence? What is the training/mean accuracy of the models corresponding to the reported test/worst-group accuracy (in Figure 6)? Lines 314-316 suggest that some hyperparameter tuning was performed, but this seems to contradict the training details (Section 5.2) which involve a fixed set of hyperparameters. How exactly were the hyperparameters tuned? Was there any early stop based on validation accuracy?

[1] Sagawa, Shiori, et al. "An investigation of why overparameterization exacerbates spurious correlations." International Conference on Machine Learning. PMLR, 2020.

**Questions:**

Please see the weaknesses.

---

### Official Review · Reviewer_APcR · 2024-11-04

**Soundness:** 2
**Presentation:** 4
**Contribution:** 3
**Rating:** 3
**Confidence:** 5

**Summary:**

This paper suggests that simple drawing of more data from the same distribution could mitigate the models' overfitting to spurious correlation within the training data. Across various types of data such as reasoning, images, and languages, authors demonstrate that training models with more (typical) data leads them to learn more robust representation against spurious correlations.

**Strengths:**

- The finding is novel; raising an interesting question that has been under-explored by previous studies.
- The paper addresses the critical research question of mitigating dataset bias (spurious correlation) within training, which is essential for reliable deployment of machine learning models especially for safety critical applications.

**Weaknesses:**

Does this observation invariantly occur when spurious correlation is **strong**? Is there a possibility that this paper's claim of "starving network will ignore the spurious features with enough data to encode the general, invariant function" is only valid when such correlation is so weak that simple drawing more data could overcome it? Regarding this,

1) This paper lacks the detailed description of the severity of correlation, and the systematic analysis (evaluation) on how invariant the observation is across different severity. Thus, additional evaluation on simply controllable dataset, e.g., ColoredMNIST, should be included.

2) How efficient and effective is simply drawing more data for debiasing, against existing debiasing strategy [1,2,3]? Does simply adding more data outperform other more complicated debiasing method?

Overall, I believe the crucial factor, severity of bias, is missing in this paper while it might significantly determine the observation in preventing models from learning debiased features.

[1] Bahng et al., Learning De-biased Representations with Biased Representations, ICML 2020 \
[2] Nam et al., Learning from Failure: Training Debiased Classifier from Biased Classifier, NeurIPS 2020 \
[3] Lee and Kim et al., Learning Debiased Representation via Disentangled Feature Augmentation, NeurIPS 2021

**Questions:**

See weakness.

---

### Official Review · Reviewer_Xu7W · 2024-11-04

**Soundness:** 1
**Presentation:** 2
**Contribution:** 1
**Rating:** 1
**Confidence:** 5

**Summary:**

The paper explores how deep neural networks often rely on spurious correlations, especially when they lack sufficient training data to encode invariant, generalizable features. Through experiments across reasoning, vision, and language tasks, the authors find that simply increasing training data while maintaining the same distribution can reduce reliance on spurious features in reasoning tasks. However, in more complex domains, such as vision and language, this approach sometimes exacerbates spurious correlations due to the presence of atypical samples. To address this, they propose a method to selectively sample typical data points, which helps networks focus on invariant features even in complex distributions.

**Strengths:**

- **Originality**: The paper introduces the concept of “invariance starvation,” a novel hypothesis suggesting that models over-rely on spurious features when they lack sufficient data to learn invariant features.
- **Quality**: The study has comprehensive tests conducted across reasoning, vision, and language tasks.
- **Clarity**: The paper is well-organized, with a clear exposition of the invariance starvation hypothesis and experimental results, enabling readers to follow the progression from hypothesis to empirical evaluation.
- **Significance**: The paper addresses an important issue in machine learning model robustness, making this work relevant to researchers and practitioners interested in improving model generalization and fairness.

**Weaknesses:**

1. The paper lacks a comprehensive review of existing work on spurious correlation mitigation, leaving out relevant recent studies. Including key papers (see Questions for some examples), would better contextualize this study’s contributions.
2. The study lacks a comparison with baseline algorithms, making it difficult to assess how well the proposed method performs relative to existing techniques for mitigating spurious correlations. Including baselines from prior work on invariant feature learning and spurious correlations would strengthen the study by providing a clearer performance benchmark.
3. The study lacks clarity regarding the design and configuration of tasks, particularly around the application of rules and dataset modifications. This lack of transparency makes the study difficult to reproduce and weakens the quality of the experimental design.
4. Many claims are unsupported by references or evidence, reducing the study's credibility. The observations drawn from experiments are not always substantiated by the figures referred to. Strengthening the argument with relevant references and ensuring experimental conclusions are well-supported by visual data would enhance the clarity and rigor of the paper.
5. Although the authors propose a method to mitigate spurious correlations, the resulting worst-group accuracy remains low both in absolute terms and in comparison to state-of-the-art methods. This suggests that spurious correlations may not be fully mitigated and raises the possibility that the method is primarily filtering out noisy examples rather than addressing spurious correlations directly. Conducting ablation studies could clarify the source of observed improvements, helping to isolate the effects of the proposed method and determine whether it genuinely mitigates spurious correlations or if other factors contribute to the results.

**Questions:**

1. Most claims about the existing literature in the introduction section are not supported by references. For example, in the statement “Of the many factors that influence the degree of spurious feature reliance, the primary factor commonly discussed in literature is that of predictive power,” which studies specifically support this assertion? Could the authors clarify what they mean by “predictive power” in this context, and how it relates to the proportion of the training set containing the spurious feature?
2. The paper does not thoroughly review recent relevant work on spurious correlations, including notable algorithms and methods published before July 1, 2024. Below are examples of studies that could enhance the related work section:
    - Wang et al. "On the Effect of Key Factors in Spurious Correlation." AISTATS 2024.
    - Yang et al. "Identifying Spurious Biases Early in Training through the Lens of Simplicity Bias." AISTATS 2024.
    - Lin et al. "Spurious Feature Diversification Improves Out-of-distribution Generalization." ICLR 2024.
    - Deng et al. "Robust Learning with Progressive Data Expansion Against Spurious Correlation." NeurIPS 2023.

    Including these and other works in a direct comparison, or explaining why they are not applicable, would provide more context and better position this study within current spurious correlation research. Yang et al. (2024) is particularly relevant, as it demonstrates how simplicity bias in model training can lead to early reliance on spurious correlations, proposing methods to detect and mitigate this bias before it leads to overfitting. Could the authors address how their approach compares to, or builds on, the early spurious bias identification methods proposed in Yang et al. (2024)?

3. What is the rationale behind selecting different training set sizes for Task 1, 2, and 3? Could the authors specify the sizes and distributions of the spurious and shifted test sets used in each task to improve transparency and reproducibility?
4. In section 3.4:
    1. What is the precise rule applied to “a portion (majority) of all samples”? For example, in Task 1, does the enforced rule state that “the variables q and p always occur together and q is equal to the negation of p,” or is it only “q is equal to the negation of p”? Could the authors clarify the exact rule and the proportion of samples affected? Similar clarifications would also be helpful for Tasks 2 and 3.
    2. The authors refer to Figure 3 (a, b, and c) to support their observations, yet these insights are not clearly depicted in the figure. Could the authors provide a more explicit visual representation or a separate explanation of these observations?
    3. The term “uniform test set” is used in this section but is not defined. Could the authors provide a description of what they mean by “uniform test set” to help readers better understand the testing methodology?
5. The CelebA setting differs from those used in prior work, making direct comparisons challenging. Could the authors explain why they chose a different setup and clarify if the MultiNLI setting aligns with that of Sagawa et al. (2020)? Furthermore, could the authors provide details on the sizes of each group in the modified CelebA and MultiNLI datasets to facilitate replication and interpretation of the results?

---

### Note · Authors · 2024-11-23

**Comment:**

We thank the reviewer for their comments.

**Withdrawal Confirmation:**

I have read and agree with the venue's withdrawal policy on behalf of myself and my co-authors.